# Pedigree-Based Description of Danubia Alba Rabbit Breed Lines

**DOI:** 10.3390/ani14182740

**Published:** 2024-09-21

**Authors:** János Posta, Csongor Demeter, Zoltán Német, Máté Sándor, Zsolt Gerencsér, Zsolt Matics

**Affiliations:** 1Department of Animal Husbandry, Institute of Animal Science, Biotechnology and Nature Conservation, Faculty of Agricultural and Food Sciences and Environmental Management, University of Debrecen, H-4032 Debrecen, Hungary; 2Cargill Feed Ltd., H-1087 Budapest, Hungary; csongor_demeter@cargill.com; 3S&K-Lap Ltd., H-2173 Kartal, Hungary; nemet.zoltan@sklap.hu (Z.N.); sandor.mate@ifarm.hu (M.S.); 4Diagnostic Center for Production Animals, Department of Pathology, University of Veterinary Sciences, H-2225 Ullo, Hungary; 5Tetrabbit Ltd., H-6500 Baja, Hungary; 6Institute of Animal Sciences, Hungarian University of Agriculture and Life Sciences, H-7400 Kaposvar, Hungary; gerencser.zsolt@uni-mate.hu; 7Department of Animal Sciences, Széchenyi István University, H-9200 Mosonmagyarovar, Hungary; matics.zsolt@ga.sze.hu

**Keywords:** pedigree quality, genetic diversity, gene origin, homozygosity, *Oryctolagus cuniculus* var. domestica

## Abstract

**Simple Summary:**

Maintenance of genetic diversity is important for commercial livestock breeds as well. The Danubia Alba rabbit is a synthetic breed; its breeding goal is meat production. The primary goal of our study was to provide information about the demographic history of Danubia Alba and support the breeder company during its selection work. Besides homozygosity, the deepness of the pedigree, the generation interval, and gene variability were evaluated. We found that the lines of genetic variability decreased, probably due to the selection work. The homozygosity of the lines was estimated using different methods, like Ballou’s, Wright’s, and Kalinowski’s inbreeding coefficients. Though current inbreeding coefficients were different among lines, the present inbreeding level came mostly from previously fixed alleles for each of them. Effective population sizes were estimated, and none of the lines was found to be critical.

**Abstract:**

The diversity of livestock animal breeds is an integral part of global biodiversity and requires careful management for sustainability and future availability. Avoiding inbreeding is a crucial aspect of mating of breeding animals. Our aims were to describe the quality of the pedigree, generation interval, gene origin, inbreeding, and effective population size of Danubia Alba rabbit lines. Line “D” is the maternal, whereas lines “C” and “X” are used as the paternal lines. The pedigree information was followed back from the actual breeding rabbits up to the founder animals. The rabbits having offspring in 2023 were chosen as reference populations for each line. The complete generation equivalent (GenCom) was 17.68 for line “C”, 18.32 for line “D”, and 17.49 for line “X”, respectively. The maximum number of generations (GenMax) was above 30 for each line. The estimated bottleneck effect is mostly the result of selection and not a real genetic loss. The Wright inbreeding coefficient (F_Wright) was the highest for the “X” line rabbits, whereas it was the lowest for the line “D”. Kalinowski’s decomposition of inbreeding showed that it originated mostly from the past; the current fixation of alleles was quite similar for the line “C” and “D”. Based on the predicted effective population sizes, it seems that there is no problem in maintaining of Danubia Alba lines.

## 1. Introduction

The diversity of livestock animal breeds is an important aspect of global biodiversity and requires careful management for sustainability and future availability. There is an emphasis on knowledge of the genetic diversity of native and endangered breeds, though it is also important to keep the long-term high production system of commercial breeds. The evaluation of pedigree data may give information about demographic changes in the population as well as from homozygosity. Avoiding inbreeding is a crucial aspect during mating of breeding animals, as mating of related animals might increase the risk of purging [1]. Effective population size is also an important measurement to take into account in mating designs [2]. Curik et al. emphasized that knowledge of inbreeding level might help to maintain genetic variance, and its effect on production data could be identified [3].

Despite being economically important and having a relatively short generation interval, only a few studies reported on rabbits’ genetic variability. Population structure and genetic diversity of Pannon White rabbits were evaluated in 2010 [4] and 2024 [5]. The closed populations of German Angora [6] and New Zealand White [7] were characterized in India. Both studies reported increasing inbreeding and low effective population sizes. Besides the small but worldwide known populations, the native rabbit breed was also evaluated [8].

Danubia Alba rabbit breed is a synthetic composite breed with three different lines. Line “D” is maternal, whereas lines “C” and “X” are used as the paternal lines. The breeding started based on the genetic basis of the Hycole breed, while the Danubia Alba breed lines have a closed studbook. No crossing is allowed for breeding animals; crossing is only used in the production system. The production level of the terminal-crossed rabbits can be seen in Table 1.

The main breeding goals are to reduce the use of antibiotics and increase resistance to environmental influences along with a high reproduction rate.

Our study focused on the evaluation of the pedigree quality, generation interval, gene origin, homozygosity, and effective population size of the Danubia Alba rabbit breed.

## 2. Materials and Methods

The pedigree information of Danubia Alba rabbit lines was supplied by the Hungarian breeding organization (Danubia Alba Nyúltenyésztő Egyesület, Kerekegyháza), responsible for the maintenance of the Danubia Alba breed. The studbook data of the registered rabbit per lines (“C”, “D” and “X”) up to 2023 were analyzed. The pedigree information was followed back from the actual breeding rabbits up to the founder animals.

The database contained the following information for each rabbit: name of the individual; name of the sire and the dam; birth date; and sex. There were the pedigree data of 63,065 from line “C”, 201,030 from line “D” and 99,143 from line “X”, respectively. The rabbits having offspring in 2023 were chosen as reference populations for each line. The actual breeding stock consisted of 678, 2580, and 606 rabbits in lines “C”, “D” and “X”, respectively.

The initial database was checked using the Pedigree Viewer 6.5 software (Armidale, Australia) [10]. The research analysis was performed using Endog 4.8 software (Madrid, Spain) [11]. The various measurements for inbreeding levels were estimated using Grain 2.2 (Wien, Austria) [12] software.

A detailed explanation of the used parameters is shown in Posta et al. (2024) [5], so here we only give the list of the numbers that were used to describe the Danubia Alba lines:Pedigree completeness was described using the maximum number of generations (GenMax), number of full generations traced (GenCom), and equivalent complete generations (GenEqu) [13];Generation interval on four different pathways [14];Number of founders (f: number of ancestors with two unknown parents);Effective number of founders (f_e_) [15];Effective number of ancestors (f_a_) [15];Number of ancestors responsible for 50%, 60%, …, 100% of the genetic variability (f_a_50, f_a_60, …, f_a_100);Inbreeding coefficient (F_X_) [16];Average relatedness (AR) [17];Ancestral inbreeding coefficients were estimated to discover whether alleles were identical by descent for the first time or whether they were already homozygous before. The following ancestral inbreeding coefficients were calculated: ancestral inbreeding coefficient according to Ballou (F__BAL_) [18], Kalinowski et al. (F__KAL_ and F__KAL_NEW_; F__KAL_NEW_ = F − F__KAL_; F is the Wright’s inbreeding coefficient, and F__KAL_ is Kalinowski et al.’s ancestral inbreeding coefficient) [19], and the ancestral history coefficient (A_HC_) [20];Effective population size based on the individual increase in inbreeding [21], increase in coancestry [22], and family size variances using the formula by Hill [23];Genetic conservation index (GCI) [24].

## 3. Results

Evaluation of pedigree quality might be important for future genetic estimations. The deeper the pedigree, the more reliable our results will be. The pedigree quality (depth and completeness) highly influences the estimates of inbreeding level [25]. Table 2 describes indexes to characterize the quality of the pedigree. Lines “C” and “X” have relatively small active populations, whilst line “D” population size is around four times larger. The equivalent complete generations varied between 23.69 and 25.35. It was the highest for line “X” and the lowest for line “C”. The average of complete generations was the highest (18.32) for line “D”; there were 56 rabbits in this reference group with 20 generations of depth pedigree where all ancestors were known. At least 15 generations were completely known for all rabbits in the three reference populations. The average maximum generations were above 30 for each breed. There were six animals in line “X” spanning 39 pedigree generations. In the active populations, every rabbit had at least one known ancestor at the 28th generation within their pedigree.

The predictions for the generation intervals are presented in Table 3. The pairwise comparison of the four pathways was performed using an independent sample *t*-test. The buck pathways were significantly longer (*p* < 0.05) compared to doe-related pathways for line “C” while there was an opposite directional difference for line “D”. The buck-to-son pathway was longer compared to other pathways for line “X”. In general, the pathways were longer for line ”C” and shorter for line “X”.

Table 4 shows parameters describing the genetic variability of the reference populations. Interestingly, N_f_ and N_a_ had the same values for the “X” line. There were huge differences among the f_e_ values; they varied between 23 and 92, whilst f_a_ values were between 18 and 58. Both values were smallest for line “X” and highest for line “D”. The existence of bottleneck effect was found for each line as the ratio of the effective number of ancestors and an effective number of founders was smaller than 1. The ratio was higher for the line “X”, whereas it was the smallest for the “C” line.

Table 5 presents the most important ancestors’ contribution to the genetic variability for the reference population. These rabbits accounted for almost 48% of the genetic variability for the “C” line, nearly 32% for the “D” line, and around 54% for the “X” line. The most important ancestor covered 4.1% of the genetic variance within the “D” line and 7.56% within the “C” line. In the “X” line, the most important ancestor accounted for 13.4% of the genetic diversity. These numbers also confirmed that the “X” line was less diverse compared to “C” and “D” lines.

Table 6 shows information from the genetic diversity of the three reference lines. The present population of line “X” could be characterized by only 112 individuals. These values were 128 and 482 in the other two lines. Only 7 and 11 animals covered half of the total genetic variability for line “C” and line “X”, respectively. In the “D” line reference population, it was higher, with 20 rabbits. The trends of these numbers are in alignment with the numbers presented in Table 5.

The three reference stocks are relatively small, so avoiding mating of related rabbits is not possible. Computation of ancestral inbreeding coefficients could provide the information that alleles became identical by descent for the first time or were already homozygous before. Table 7 summarizes the information about homozygosity of the three lines. It is not usual that all rabbits in the reference populations were inbred. The inbreeding was calculated in several different ways. Besides the classical inbreeding coefficient, ancestral inbreeding coefficients were also estimated to determine if inbreeding has happened presently or in the past. The Wright inbreeding coefficient was the highest for the active “X” line rabbits. The inbreeding coefficient usually increases over time, especially in small and closed populations where the mating of related individuals is unavoidable. The lowest inbreeding was estimated for line “D”. The Ballou’s ancestral inbreeding coefficient and the A_HC_ were higher than other estimated parameters; however, their tendency was similar to Wright’s coefficient. The estimated F__Kal_new_ values were very similar for each line. These values were smaller than F__Kal_, so inbreeding originated mostly from the past. Kalinowski’s inbreeding coefficients were the lowest for line “D”.

The effective population size is a very important parameter not only for endangered animals but industrial breeds as well. As the breed has a closed stud book and crossing among lines is also not possible, loss of genetic variability is not avoidable. Table 8 shows the different Ne values, which were estimated in three different ways. The contribution of breeding individuals to the next generation is not equal, which results in a smaller effective population size compared to the exact population size. The inbreeding-based effective population size (Ne_f) was much higher compared to the other parameters for each line. It was the lowest (85.49) for the line “X” compared to the other two lines. The effective population sizes estimated using regression procedures (Ne_reg, Ne_log) were quite similar to each other for each line. In agreement with the inbreeding-based effective population size, line “X” had the lowest coefficients while the highest values were estimated for line “D”. Estimated numbers of line “C” were between the above-mentioned two but closer to line “X”.

The annual trend of effective population size was also calculated based on the family size variances and is presented in Figure 1. The annual estimated effective population size was above 150 and fluctuated around 200 for line “D” in the last fifteen years. This parameter was below 150 for line “C” and line “X”. The annual effective population sizes changed frequently across years; the heterogeneity of lines “D” and “X” were quite similar and were above 30%, whereas the heterogeneity of line “C” was somewhat lower.

The distribution of the genetic conservation index (GCI) is presented in Table 9. The highest GCI values were estimated for the “C” line, whereas values were lowest for line “X”. Rabbits having more than 40 GCI belonged to line “C”. Interestingly, indexes were higher for line “D” than line “X”.

## 4. Discussion

The complete generations were higher for each Danubia Alba line compared to German Angora [6] and New Zealand White [7] rabbits. Our estimations highly exceeded the information from Pannon White rabbits [5]. We also estimated higher values for equivalent complete generations for each line compared to previous publications [5,6,7]. The maximum number of generations was also higher than was reported for German Angora [6] and New Zealand White [7]; however, the value in the case of the Pannon White rabbit [5] was slightly higher than in our findings. The received high numbers for pedigree quality suggest that the further estimated values will be reliable.

Interestingly, our estimation for the generation intervals was smaller compared to previous studies. The German Angora [6] and New Zealand White [7] breeding rabbits were at least three months older than the Danubia Alba rabbits, whereas Pannon White rabbits [5] were also slightly older at the birth of their offspring, which were selected as breeding candidates. The younger bucks in the breeding stock of the Danubia Alba lines allow for a larger selection response and quicker selection progress. The annual replacement rate is 120% in the breeding population, so the long-term quick replacement of breeding animals gives the risk of increased inbreeding levels within each line.

Our calculated f_a_ and f_e_ values were quite variable but larger than what was published for New Zealand White rabbits [7]. Values for line “X” were quite similar to German Angora [6], whereas lines “C” and “D” had higher f_a_ and f_e_ values. Estimations for line “C” were quite close to the values reported for the Pannon White rabbit [5]. However, the variable values were more similar to the estimations for other species. Line “C” values were close to Campolina [26] and Lippizaner [27] horses, whereas estimations for line “D” were in agreement with Posavac horse [28] and Belgian Landrace pigs [29]. Despite similar actual breeding stock, the estimated effective number of founders and ancestors for line “C” and line “X” were very different. The size of the present breeding stock and the estimated values seem to be somewhat related as the line “D” had larger breeding stock, and the estimated values were larger as well.

The ratio of f_a_ and f_e_ values reported a reasonable bottleneck effect for each line. This finding could also suggest genetic drift within the population. The strong selection of bucks could be the reason for the obtained result. Our ratios were lower compared to those values reported for German Angora [6] and New Zealand White [7] rabbits. The received value for line “X” was in agreement with the estimation for the Pannon White rabbit [4,5]. The bottleneck effect should be avoided in the case of endangered and small populations, but it is inevitable for commercial breeds where there is the expectation of changing allele and genotype frequency within the populations. From this point of view, it might be better to talk about the effect of selection and not genetic loss for commercial populations.

The huge values of the contribution of the ancestors to the gene pool for line “X” suggest that this line is mostly based on a few ancestors, which requires special attention to managing the homozygosity level within the line. The concentration of genetic variability differed across the Danubia Alba lines, though the tendencies for f_a_50, f_a_60, and f_a_70 were very similar. However, line “D” was the most diverse as the final number was almost four times higher than for line “C” and line “X”.

The inbreeding level signs the homozygosity and genetic variability of the populations. All rabbits of the current breeding population were inbred. The estimated inbreeding level, in general, strongly depended on the actual population size (reference population), as values were lower for line ”D” and higher for lines “C” and “X”. The average relatedness was near 50% higher than the Wright inbreeding coefficient, so mating of related individuals was quite frequent in each Danubia Alba line. The F_X_ was a little bit higher than 5% for line “D”, whereas it was close to 10% for line “C” and line “X”. The values for lines “C” and “X” were in alignment with the homozygosity of the Pannon White rabbit [4,5] and Ibicenco rabbit [2]. Our estimations for each line were somewhat lower compared to the mean inbreeding of the German Angora [6] and New Zealand White [7] populations. Compared to horse breeds having closed studbooks, huge similarities were found. Interestingly, the inbreeding levels of lines “C” and “X” were quite close to the values reported for the joint evaluation of several Lippizaner stocks [27,30], whereas line “D” was in alignment with the values of the Lusitano horses [31]. The estimated values for average relatedness for the different Danubia Alba lines were lower compared to New Zealand White [7]. Results for German Angora [6] were lower than the average value for line “X” but higher compared to line “C” and line “D”. However, our estimates exceeded the average relatedness of Pannon White rabbits [4,5].

The decomposition of inbreeding based on Kalinowski’s and Kalinowski’s new formula is becoming more widely used and evaluated for livestock breeds. By definition, when the F_X_ inbreeding coefficient is zero, the F__Kal_ is also zero. Despite the similar pedigree quality, the ancestral proportion of the inbreeding level was very different among the Danubia Alba lines. However, the current part of the inbreeding (F__Kal_new_) was quite similar, from 1.12 to 1.72, for each line. This suggests that the selection and mating strategy of the breed result in similar changes in inbreeding despite the different sizes of the breeding populations across the lines. Surprisingly, the F__Ballou_ and A_HC_ values highly differed among the lines, as our estimations for line “X” were quite similar to those of the Pannon White rabbit [5], whereas the results for line “C” and especially for line “D” were smaller.

Inbreeding is usually higher in the case of long pedigree information, so an effective population size could also give useful information about the homozygosity of the population. That is the reason for the suitability of effective population size to characterize the diversity of not only native breeds but commercial livestock breeds as well [32]. The low effective population size increases the risk of extinction [33]. Previous studies were supposed to have a threshold value, so a lower value suggests problems with the long-term maintenance of the given breeding stock [34]. The trend of the different estimations was similar, and the tendency was strengthened with the annual presentation of the effective population sizes. Our results confirmed that the effective population size could depend on the inbreeding level of the population. The reference population of line “D” was the highest, and the effective population size of line “D” was more than two times higher compared to estimations for line “C” and line “X”. Our estimations exceed the numbers found in previous studies for German Angora [6], New Zealand White [7], and Ibicenco [2] rabbit breeds. The effective population size of the present Pannon White population [5] is similar to Danubia Alba’s lines “C” and “X”, whereas our estimated value for line “D” was reasonably higher. The high fluctuation for line “D” between 2001 and 2010 might be because of differences in population size in those times. Our values suggest that none of the lines are endangered by genetic factors.

The higher GCI values emphasize the importance of the breeding animals. The distribution of GCI in our present study was different across lines. All values were higher than 20%, which is higher than was reported for German Angora [6] and New Zealand White [7] rabbits. The frequencies for line “D” and “X” were within the distribution estimated for Pannon White [5], whereas the current breeding stock of line “C” had higher values. There were only slight differences among the values within lines, which also strengthened the homogeneity of the breeding population. Despite the selection, the low within-line variance of GCI suggests that the contribution of founder ancestors to the actual population is quite balanced within each line.

## 5. Conclusions

The population structure of the synthetic Danubia Alba rabbit was evaluated from a pedigree information point of view, which presented the genetic structure of this commercial breed. The high values for pedigree quality allow for reliable estimations of pedigree-based population genetic parameters for the actual breeding stock. The bottleneck effect was proven for each line; the decrease in genetic variability might be due to strong selection within the lines. The evaluation of genetic drift might be a topic for future genomic analyses. The reference populations were completely inbred, though effective population sizes are not critical. It is crucial to periodically monitor inbreeding and the average relatedness to avoid possible negative effects on fitness and production traits. The periodically estimated average relatedness might be taken into account during the development of mating plans. The present inbreeding level was mostly the result of previously fixed alleles for each line.

## Figures and Tables

**Figure 1 animals-14-02740-f001:**
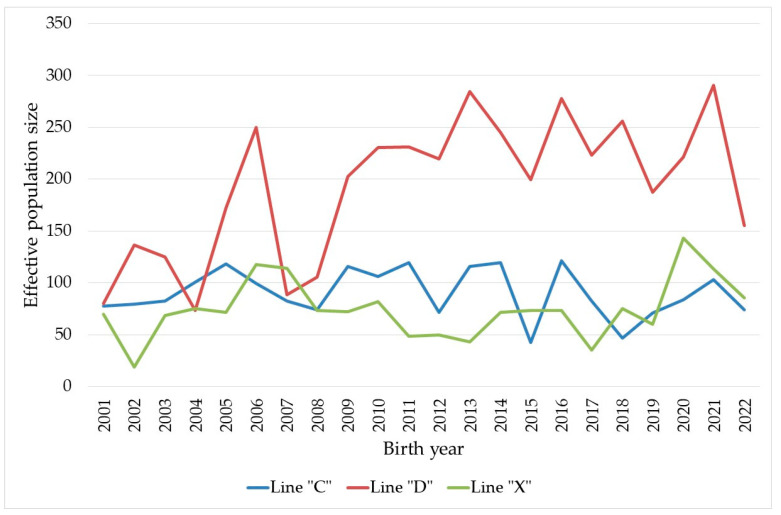
Annual trend for effective population size in Danubia Alba rabbit lines.

**Table 1 animals-14-02740-t001:** Production of the terminal Danubia Alba rabbits (mean ± standard deviation).

Gender	Weight at 5 Weeks Old	Weight at 10 Weeks Old	Weight Gain during Fattening
Does	1160 ± 147.1	2741 ± 216.2	45 ± 4.6
Bucks	1147 ± 146.9	2627 ± 223.3	42 ± 4.8

Posta-Juráskó (2022) [9].

**Table 2 animals-14-02740-t002:** Pedigree completeness in the Danubia Alba reference populations (mean ± standard deviation).

	Line “C”	Line “D”	Line “X”
GenEqu	23.69 ± 0.610	24.35 ± 0.579	25.35 ± 0.708
GenCom	17.68 ± 0.609	18.32 ± 0.706	17.49 ± 0.715
GenMax	30.67 ± 0.808	30.87 ± 0.732	36.84 ± 0.928

GenEqu: equivalent complete generations; GenCom: number of full generations traced; GenMax: maximum number of generations.

**Table 3 animals-14-02740-t003:** Estimated generation intervals for the Danubia Alba reference populations (years).

Pathways	Line “C”	Line “D”	Line “X”
Buck-to-son	0.99 (*n* = 1601) ^a^	0.77 (*n* = 3157) ^c^	0.88 (*n* = 1318) ^a^
Buck-to-daughter	1.00 (*n* = 6431) ^a^	0.77 (*n* = 34,642) ^c^	0.83 (*n* = 8280) ^b^
Doe-to-son	0.96 (*n* = 1601) ^b^	0.97 (*n* = 3152) ^a^	0.83 (*n* = 1318) ^b^
Doe-to-daughter	0.93 (*n* = 6431) ^c^	0.94 (*n* = 34,648) ^b^	0.82 (*n* = 8281) ^b^
Average	0.97 (*n* = 16,064)	0.85 (*n* = 75,599)	0.83 (*n* = 16,064)

Different superscript letters show significant differences (*p* < 0.05).

**Table 4 animals-14-02740-t004:** Founders and ancestors in the Danubia Alba reference populations.

	Line “C”	Line “D”	Line “X”
N_f_	136	305	112
N_a_	128	482	112
f_e_	55	92	23
f_a_	31	58	18
f_a_/f_e_	0.56	0.63	0.78

N_f_: number of founders; N_a_: number of ancestors; f_e_: effective number of founders; f_a_: effective number of ancestors.

**Table 5 animals-14-02740-t005:** Contribution of the ancestors to the gene pool of the Danubia Alba reference populations (%).

	Line “C”	Line “D”	Line “X”
first ancestor	7.56	4.10	13.40
second ancestor	6.75	3.81	10.62
third ancestor	6.02	3.66	6.98
first 10 ancestors	47.88	31.65	63.68

**Table 6 animals-14-02740-t006:** Concentration of genetic variability in the Danubia Alba reference populations.

	Line “C”	Line “D”	Line “X”
f_a_50	11	20	7
f_a_60	15	28	9
f_a_70	21	38	13
f_a_80	28	54	19
f_a_90	42	82	30
f_a_100	128	482	112

f_a_50; f_a_60; …; f_a_100: number of ancestors responsible for 50%; 60%; …; 100% of the genetic variability.

**Table 7 animals-14-02740-t007:** Description of homozygosity in the Danubia Alba reference populations.

	Line “C”	Line “D”	Line “X”
Inbred animals (%)	100	100	100
AR	13.30	8.84	18.73
F_X_	9.26	5.28	12.83
F__Ballou_	53.23	38.73	69.29
F__Kal_	7.51	3.66	11.70
F__Kal_new_	1.75	1.62	1.12
A_HC_	89.40	55.34	155.27

AR: average relatedness; F_X_: inbreeding coefficient; F__Ballou_: Ballou’s formula for ancestral inbreeding; F__Kal_: identical alleles were inbred in the past; F__Kal_new_: identical alleles were inbred in recent generations; A_HC_: ancestral history coefficient.

**Table 8 animals-14-02740-t008:** Effective population size in the Danubia Alba reference populations.

	Line “C”	Line “D”	Line “X”
Ne_f	122.17	321.09	85.49
Ne_reg	84.85	182.69	73.36
Ne_log	85.08	183.06	73.03

Ne_f = effective population size computed using individual increase in inbreeding; Ne_reg = effective population size computed using the regression on equivalent generations; Ne_log = effective population size computed based on the log regression on equivalent generations.

**Table 9 animals-14-02740-t009:** Frequency distribution in the Danubia Alba reference populations (%).

GCI	Line “C”	Line “D”	Line “X”
20.00 or less	0% (*n* = 0)	0% (*n* = 0)	0% (*n* = 0)
20.01 to 40.00	0% (*n* = 0)	100% (*n* = 2580)	100% (*n* = 606)
40.01 or more	100% (*n* = 678)	0% (*n* = 0)	0% (*n* = 0)

## Data Availability

The data presented in this study are available upon request from the corresponding author. The data are not publicly available due to privacy restrictions.

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
