# Peer review of "Pedigree-Based Description of Danubia Alba Rabbit Breed Lines"

_animals, 2024, doi:10.3390/ani14182740_

Round 1

Reviewer 1 Report

Comments and Suggestions for Authors

General comments:

Manuscript in the scope of the journal Animals. Interesting topic about demography and genealogy studies in rabbits, something that is scarce in the available scientific bibliography.

However, it needs some revision, especially in the Material and Methods section, which needs to be more comprehensive, clear and better explained.

A greater critical spirit is needed in the explanation of the results obtained.

Specific comments:

L26-27 - sentence with bad meaning, avoid using terms like "thing". More scientific language needed. Clarify

L28 - "... these three separate lines..." Which ones? Mention them because they are never indicated in the abstract.

The abstract lacks specific indication of some values ​​obtained, such as Ne, Fx, etc.

Key words - consider putting the rabbit's name in Latin

L41-42 - full repetition in the 1st sentence of the abstract. Change, review

L46 "thing" again - consider "aspect"...

L48 "... size IS also an important...."

L59 - "... these 3 separate lines..." Which ones? Never refered until the moment.

Only mentioned the lines in M&M section. Refer earlier.

L72 - number of records for reference pops?

L74 - "using" 2x

L81 - and if only one ancestor is known?

L88-89 - sentence not clear. Rephrase.

Globally too short M&M section. Complete.

L102 - lacking a verb in sentence

Table 1 - estimations of SD?  Lacking the legend for abbreviations in all tables presented

L120 - insert always a space/paragraph after each table.

Attention to the formatting of the acronyms Nf, Na, fe fa putting the letters underscript

L150 - sentence not clear, rephrase

L178 - bad english, rephrase

Figure 1 - Possible explanation for the huge fluctuation in values ​​in Line D?

L184 - conservation and not conversation

L184-187 - Possible explanation for the interpretation of GCI values with different rankings for the 3 lines when comparing with other parameters obtained?

Table 8 - besides the "n" you could also refer to % considering so different sizes of samples in lines

L192 - consider change "received" to estimated

L203-204 - sentence not clear. Rephrase. 

L205 - What is the average replacement rate?

L223-224 - Possible explanation for this statement?

L299-304 - nonsense information

Comments on the Quality of English Language

Needing some review of technical terminology and agreement of verbs and clarity in the definitions presented

Author Response

Response to Reviewer 1 Comments

General comments:

Manuscript in the scope of the journal Animals. Interesting topic about demography and genealogy studies in rabbits, something that is scarce in the available scientific bibliography.

Thank you for your opinion.

However, it needs some revision, especially in the Material and Methods section, which needs to be more comprehensive, clear and better explained.

We agree with the reviewer that the detailed definition of each parameter might be necessary. Besides our previous work (Posta et al., 2024 https://doi.org/10.1016/j.livsci.2024.105460), there are several other studies describing these parameters for other species. When we defined each parameter, the similarity index for the manuscript was very high. To resolve this problem, we rephrased the sentence before the listing of the shown estimations as: “The detailed explanation of the used parameters is shown in Posta et al. (2024) [4], so here we only give the list of the numbers which were used to describe the Danubia Alba lines:

A greater critical spirit is needed in the explanation of the results obtained.

Some sentences were added to the Discussion part to explain results and make some comparisons among them.

Specific comments:

L26-27 - sentence with bad meaning, avoid using terms like "thing". More scientific language needed. Clarify

Sentence was corrected.

L28 - "... these three separate lines..." Which ones? Mention them because they are never indicated in the abstract.

Explanation of Danubia Alba rabbit lines was inserted to the Abstract.

The abstract lacks specific indication of some values obtained, such as Ne, Fx, etc.

Abbreviation of parameters were added to the Abstract where it was possible.

Key words - consider putting the rabbit's name in Latin

It was added.

L41-42 - full repetition in the 1st sentence of the abstract. Change, review

The sentence was rephrased.

L46 "thing" again - consider "aspect"...

Corrected.

L48 "... size IS also an important...."

Corrected.

L59 - "... these 3 separate lines..." Which ones? Never refered until the moment.

Only mentioned the lines in M&M section. Refer earlier.

The description of the breed was moved from M&M section to the Introduction part and was extended.

L72 - number of records for reference pops?

The number of records (size of actual breeding stock) was moved from Table 1 to M&M section.

L74 - "using" 2x

Corrected.

L81 - and if only one ancestor is known?

In that case the given individual cannot be handled as founder animal.

L88-89 - sentence not clear. Rephrase.

The sentence was rephrased.

Globally too short M&M section. Complete.

The M&M section was slightly changed. We completely agree with the reviewer that the detailed definition of each parameter might be necessary and presented in M&M section. When we defined each parameter, the similarity index (eg. iThenticate or Turnitin reports) for the manuscript was very high. Besides our previous work (Posta et al., 2024 https://doi.org/10.1016/j.livsci.2024.105460), there are several other studies describing these parameters for other species. To resolve this problem, we rephrased the sentence before the listing of the shown estimations as: “The detailed explanation of the used parameters is shown in Posta et al. (2024) [4], so here we only give the list of the numbers which were used to describe the Danubia Alba lines:

L102 - lacking a verb in sentence

The sentence was rephrased.

Table 1 - estimations of SD?  Lacking the legend for abbreviations in all tables presented

SD was added to the table. Abbreviations were inserted to all tables where it was applicable.

L120 - insert always a space/paragraph after each table.

Corrected.

Attention to the formatting of the acronyms Nf, Na, fe fa putting the letters underscript

Corrected.

L150 - sentence not clear, rephrase

The sentence was rephrased.

L178 - bad english, rephrase

The sentence was split to two sentences and was rephrased.

Figure 1 - Possible explanation for the huge fluctuation in values in Line D?

The high fluctuation for line “D” between 2001 and 2010 might be because of differences in population size in those times.

L184 - conservation and not conversation

Corrected.

L184-187 - Possible explanation for the interpretation of GCI values with different rankings for the 3 lines when comparing with other parameters obtained?

A few explanation added to the Discussion part of the manuscript.

Table 8 - besides the "n" you could also refer to % considering so different sizes of samples in lines

The heading of table 8 was changed as ‘GCI’ was added. Besides the ‘%’ n the title of the table, it was added to each cells.

L192 - consider change "received" to estimated

Corrected.

L203-204 - sentence not clear. Rephrase.

The sentence was removed as the information about the closed studbook of the lines was written previously.

L205 - What is the average replacement rate?

Usually there are six kindlings per year, the annual replacement rate is 120%. It was added to the rephrased sentence.

L223-224 - Possible explanation for this statement?

In small endangered populations and native breeds, there is the demand for keep and maintain all alleles and genotypes to maximize genetic variability, so no selection within the population. For commercial breeds, the primary aim is to increase the production, so there is selection for the desired traits. This selection will result that the frequency of favourable alleles/genotypes will increase in the population, while frequency of other alleles will decrease or they will disappear from the population.

The sentence was slightly modified.

L299-304 - nonsense information

Thank you, it was corrected.

Reviewer 2 Report

Comments and Suggestions for Authors

The description of the material is very poor. There are three lines of Danubia rabbit, but what is the origin of every line, how genetically do they differ, what is the population of every line, are they open or closed, are there any crossbreeding between lines?

The history of line is influencing genetic parameters calculated from pedigree data.

Inline 77 is the sentence, “The following numbers were used to describe the Danubia Alba lines,” and later, it is a long list of parameters that were calculated on pedigree data. The word “numbers” fits the paper because after reading it, there are many numbers but no explanation and discussion about their value or connection between parameters.

In my opinion, the paper presents the ability of authors to calculate many parameters, but there is a lack of connection with the implementation of these values.

Many parameters are connected, and the discussion is very much needed, line number of founders explaining some percent of the variability. Why is there a difference between the lines? The history of creating those lines could explain a part.

There is no explanation for the differences between different inbreeding values. Why the relations between parameters are not the same within lines?

In Figure 1, there is a presented trend in effective population size – but which one from Table 7?

So, there are plenty of results, but what does it mean for rabbits and future breeding?

Author Response

Response to Reviewer 2 Comments

The description of the material is very poor. There are three lines of Danubia rabbit, but what is the origin of every line, how genetically do they differ, what is the population of every line, are they open or closed, are there any crossbreeding between lines?

The description of the breed was moved from M&M section to the Introduction part and was extended with relevant information.

The history of line is influencing genetic parameters calculated from pedigree data.

Completely agree with Reviewer. That is the reason to show 20 years long history what seems to be short for humans but more than 20 generation for rabbits.

In line 77 is the sentence, “The following numbers were used to describe the Danubia Alba lines,” and later, it is a long list of parameters that were calculated on pedigree data. The word “numbers” fits the paper because after reading it, there are many numbers but no explanation and discussion about their value or connection between parameters.

We tried to add some explanation to Result and Discussion sections to improve the manuscript.

In my opinion, the paper presents the ability of authors to calculate many parameters, but there is a lack of connection with the implementation of these values.

Some sentences were added to Discussion part to improve it.

Many parameters are connected, and the discussion is very much needed, line number of founders explaining some percent of the variability. Why is there a difference between the lines? The history of creating those lines could explain a part.

Some explanation of the breed and lines was added to the Introduction section.

There is no explanation for the differences between different inbreeding values. Why the relations between parameters are not the same within lines?

Some explanation was added to the Discussion section.

In Figure 1, there is a presented trend in effective population size – but which one from Table 7?

The effective population size shown in Figure 1 was calculated based on family size variances. This information was inserted to Materials and Methods section and also shown in Results section.

So, there are plenty of results, but what does it mean for rabbits and future breeding?

Conclusion part was modified with possible future topics based on our results.

Round 2

Reviewer 1 Report

Comments and Suggestions for Authors

Thanks for the revision.

Good work in the improvement of the paper.

No further comments

Comments on the Quality of English Language

Could be improved, still with some sentences lacking better structure.

Author Response

Thank you for your opinion.